# Adult K18-hACE2 mice are suitable for studying intranasal SARS-CoV-2 infection but not direct-contact transmission

Jiseon Kim,[1] Sung-Hee Kim,[1] Donghun Jeon,[1] Haengdueng Jeong,[2] Chanyang Uhm,[1] Heeju Oh,[1] Kyungrae Cho,[3] Yejin Cho,[1] Sumin Hur,[1] In Ho Park,[1,4] Jooyeon Oh,[5] Jeong Jin Kim,[1] Jun-Young Seo,[1] Jeon-Soo Shin,[1,4,5] Je Kyung Seong,[6,7,8,9] Ki Taek Nam[1]

**ABSTRACT** The rapid global spread of severe acute respiratory syndrome coronavirus-2 (SARS-CoV-2) since 2019 emphasizes the need to understand its transmission routes, which mainly comprise airborne and contact transmission. Contact transmission, where the virus spreads through direct or indirect contact, is key to the disease epidemiology. Therefore, investigating contact transmission in animal models is crucial for understanding SARS-CoV-2 behavior and developing effective preventive measures. Although ferrets, cats, and hamsters have been established as models for studying contact transmission, the susceptibility of mice (the most commonly used experimental animal model) to SARS-CoV-2 contact infection remains uncertain. In this study, we investigated whether SARS-CoV-2 can spread via contact transmission in adult K18-hACE2 mice with different genetic backgrounds, including those with mitomycin C-induced immunodeficiency. We conducted contact-transmission experiments by co-housing K18-hACE2 mice intranasally infected with SARS-CoV-2 S type (isolated in Korea) alongside uninfected adult K18-hACE2 mice. Mice with genetically different backgrounds subjected to contact infection exhibited no changes in clinical signs or histopathological changes in the respiratory tract and extrapulmonary organs. Additionally, neither SARS-CoV-2 nor neutralizing antibodies were detected in any of the tested samples. Their immune responses remained unchanged, and contact transmission was not observed, even in immunodeficient mice. Collectively, these findings suggest that adult K18-hACE2 mice are not susceptible to contact infection with SARS-CoV-2, highlighting the role of immune mechanisms in viral spread and the limitations of this model for studying human transmission pathways. Our results underscore the importance of utilizing appropriate animal models to accurately elucidate transmission dynamics.

**IMPORTANCE** Understanding the mechanisms of severe acute respiratory syndrome coronavirus-2 infection and transmission is essential for preventing and treating coronavirus disease 2019. Varying opinions exist regarding the occurrence of contact infection in mice. Here, we aimed to induce contact infection under various conditions in K18-hACE2 mice. By measuring clinical symptoms, viral loads, and neutralizing-antibody titers and conducting pathological analyses, we demonstrated that contact infection did not occur in K18-hACE2 mice. These findings underscore the importance of selecting appropriate experimental animal models to guide future studies on viral infections.

**KEYWORDS** SARS-CoV-2, contact transmission, hCoV-19/Korea/KCDC03/2020_WA-1, K18-hACE2 mice model, histopathological change, mouse background, immune response

S evere acute respiratory syndrome coronavirus-2 (SARS-CoV-2) has rapidly spread worldwide since its initial emergence in 2019, resulting in a pandemic. The rapid spread of SARS-CoV-2 has underscored the importance of understanding its primary

Address correspondence to Je Kyung Seong, snumouse@snu.ac.kr, or Ki Taek Nam, kitaek@yuhs.ac.

Jiseon Kim and Sung-Hee Kim contributed equally to this article. The author order was determined based on the amount of contribution to the experimental design.

The authors declare no conflict of interest.

See the funding table on p. 15.

transmission routes, which predominantly include airborne dissemination and direct contact (1–6). The canonical receptor for SARS-CoV-2, known as angiotensin-converting enzyme 2 (ACE2), is expressed not only in respiratory organs (such as the oral and nasal mucosa and lungs) but also in digestive organs (like the stomach, small intestine, and colon). ACE2 is also expressed in immune system organs (like the lymph nodes, thymus, bone marrow, and spleen) and in the brain, thus enabling systemic infection (7, 8). In addition to ACE2, alternative receptors might bind with SARS-CoV-2, including neuropilin-1, CD147, CD209L, asialoglycoprotein receptor 1, kringle containing transmembrane protein 1, and glucose-regulated protein 78 (9–13). In humans, SARS-CoV-2 detection in feces or urine has highlighted the potential for indirect transmission, augmenting concerns about escalating infection rates (14–16).

Animal models are invaluable tools for exploring complex transmission dynamics (17). Representative contact-transmission models include ferrets, hamsters, and cats (18). In ferrets, infection has been confirmed to spread through direct contact (19). Furthermore, some data have shown that viruses can be detected in the saliva, urine, and feces of infected ferrets, suggesting that noncontact transmission can also occur (20). In addition, direct-contact infection has been reported in cats co-housed with other SARS-CoV-2-infected cats, where the virus was detected in the nasal and rectal swabs of the co-housed cats (21, 22). Previously, hamsters showed induction of upper respiratory-tract lesions similar to those in humans and demonstrated SARS-CoV-2 contact transmission (23–25). In those studies, the virus was detected in the skin and feces of the infected hamsters, and clinical signs of infection, such as weight loss, were observed. Additionally, transmission occurred when uninfected hamsters were housed with hamsters infected with SARS-CoV-2, and those animals subsequently died (26).

As one of the most widely used experimental animal models, mice are frequently employed in studies of respiratory infections and disease pathogenesis. Techniques such as oral or direct intranasal administration and inhalation are commonly used to introduce pathogens, enabling controlled infection and the examination of disease progression (27, 28). Numerous studies on SARS-CoV-2 have also utilized mice to investigate viral-infection mechanisms, immune responses, and potential therapeutic strategies, but SARS-CoV-2 signs do not develop in wild-type mice (29–31). Although some data have shown that infection can occur in severe combined immunodeficiency (SCID) mice infected with a beta variant of SARS-CoV-2, the mice did not exhibit signs of viral replication in their lungs after infection, and their lung lesions gradually improved over time (32). Therefore, experiments have been conducted using genetically modified mice engineered to express receptors capable of binding to SARS-CoV-2 or the receptor-binding domain of the SARS-CoV-2 spike (S) protein variant modified to bind mouse ACE2 with high affinity, thus rendering mice highly susceptible to SARS-CoV-2 infection (16, 33–37). In genetically modified mice, lesions were observed in the lungs and other organs, similar to those seen in humans (15, 38, 39).

While direct infection through the diet or nasal routes has been confirmed in genetically modified mice, contact transmission has been debated (37, 40–42). One research group reported rare viral detection in mice after several days of contact with infected human angiotensin-converting enzyme 2 (hACE2)-transgenic mice, although viral detection in the lungs (the main target organ) was not confirmed (41). Moreover, in keratin 18 (*K18*) promoter-derived, hACE2-transgenic mice, researchers observed that contact infection occurred readily during the neonatal stage. However, in adult mice, the virus was detected in only one mouse, with no virus detected in the others (43). In another study, the researchers introduced the N501Y mutation into the receptor-binding domain of SARS-CoV-2 and used it to induce SARS-CoV-2 contact infection in wild-type mice. They reported that contact infection occurred only with the beta variant (B.1.351), which was isolated from a South African traveler (37, 44). However, another report demonstrated that when the same viral strain was used to study contact infection in wild-type mice, no transmission occurred (42). These results highlight the uncertainty surrounding the induction of contact infection in mice.

In this study, we aimed to evaluate SARS-CoV-2 contact infection in adult K18-hACE2 mice with two different backgrounds. We used K18-hACE2 mice with the commonly used C57BL/6 background and K18-hACE2 mice with the FVB/NJ background (developed in Korea) to assess contact transmission (45). We housed naïve K18-hACE2 mice with K18-hACE2 mice infected with S-type SARS-CoV-2 (isolated in Korea) at both the early and late stages of infection. We conducted a comprehensive analysis on the mice subjected to contact transmission, including clinical-symptom evaluation, pathological analysis, and virus detection in respiratory organs. Our pathological analysis was extended to non-respiratory organs to accurately assess whether contact transmission occurred. Lastly, we studied contact infection in K18-hACE2 mice with mitomycin C (MMC)-induced immunodeficiency, which confirmed the necessity of immune mechanisms for contact infection.

## RESULTS

### Clinical features and pathogenesis after intranasal SARS-CoV-2 infection in K18-hACE2 mice

We used K18-hACE2 mice to establish an animal model of SARS-CoV-2 infection. *In situ* hybridization confirmed that hACE2 was expressed in the alveoli, bronchi, and vessels of K18-hACE2 mouse lungs and in the trachea (Fig. S1A through C). To assess clinical signs and lung pathogenesis, SARS-CoV-2 was intranasally administered to K18-hACE2 mice, and observations were made at different days post-infection (dpi). The body temperature and weights of the SARS-CoV-2-infected K18-hACE2 mice decreased after 2 dpi. By 7 dpi, the infected mice exhibited significant reductions, with a >10°C decrease in body temperature and a 20% decrease in body weight. In contrast, mock-infected K18-hACE2 mice did not show any changes (Fig. 1A and B). SARS-CoV-2-infected mice began dying at 5 dpi, with only 20% surviving at 7 dpi (Fig. 1C), as reported previously (46, 47). Next, we compared the compositions of white blood cells in the peripheral blood. The neutrophil:lymphocyte ratio (a diagnostic and prognostic marker of clinical outcomes) increased significantly, depending on the days post-infection (Fig. 1D) (48, 49).

Plaque assays were performed to compare viral titers in lungs from SARS-CoV-2-infected mice at multiple time points with those from mock-infected mice. SARS-CoV-2 PFUs were highest at 2 dpi and decreased by 7 dpi (Fig. 1E; Fig. S2A). Pathological differences occurred in the SARS-CoV-2-infected mice over time. In the lungs at 2 dpi, immune cells infiltrated the alveolar region through blood vessels, and vascular edema and capillary dilation were confirmed. At 7 dpi, the lesions worsened, and the pathological scores increased significantly (Fig. 1F and G). We performed immunohistochemical (IHC) staining for the SARS-CoV-2 nucleocapsid (N) protein to verify viral distributions in the lungs of SARS-CoV-2-infected mice. Over 70% of the lung areas were positive for the N protein at 2 dpi, but by 7 dpi, the percentage of infected areas had significantly decreased (Fig. 1H and I), which correlated with the PFU data for the lungs. The absence of *S* gene detection in the lungs and trachea of the recipient group through *in situ* hybridization further confirmed that contact infection did not occur in the respiratory system of adult K18-hACE2 mice (Fig. S3A and B).

We also performed a pathological analysis of extrapulmonary organs. In the spleen, white blood cell apoptosis (associated with the occurrence of a cytokine storm) was observed at 2 dpi, with extensive damage at 7 dpi (Fig. S4A and B) (38). In the small intestine, goblet cell hyperplasia, villous atrophy, and villous necrosis were only observed at 7 dpi (Fig. S4D and E). Similarly, multifocal perivascular cuffing in the brain was only detected at 7 dpi (Fig. S4G and H) (15, 50). Consistent with previous research findings, our data confirmed that K18-hACE2 mice intranasally infected with SARS-CoV-2 exhibited pathological signs in their pulmonary and extrapulmonary organs, leading to systemic infection and eventual mortality (35).

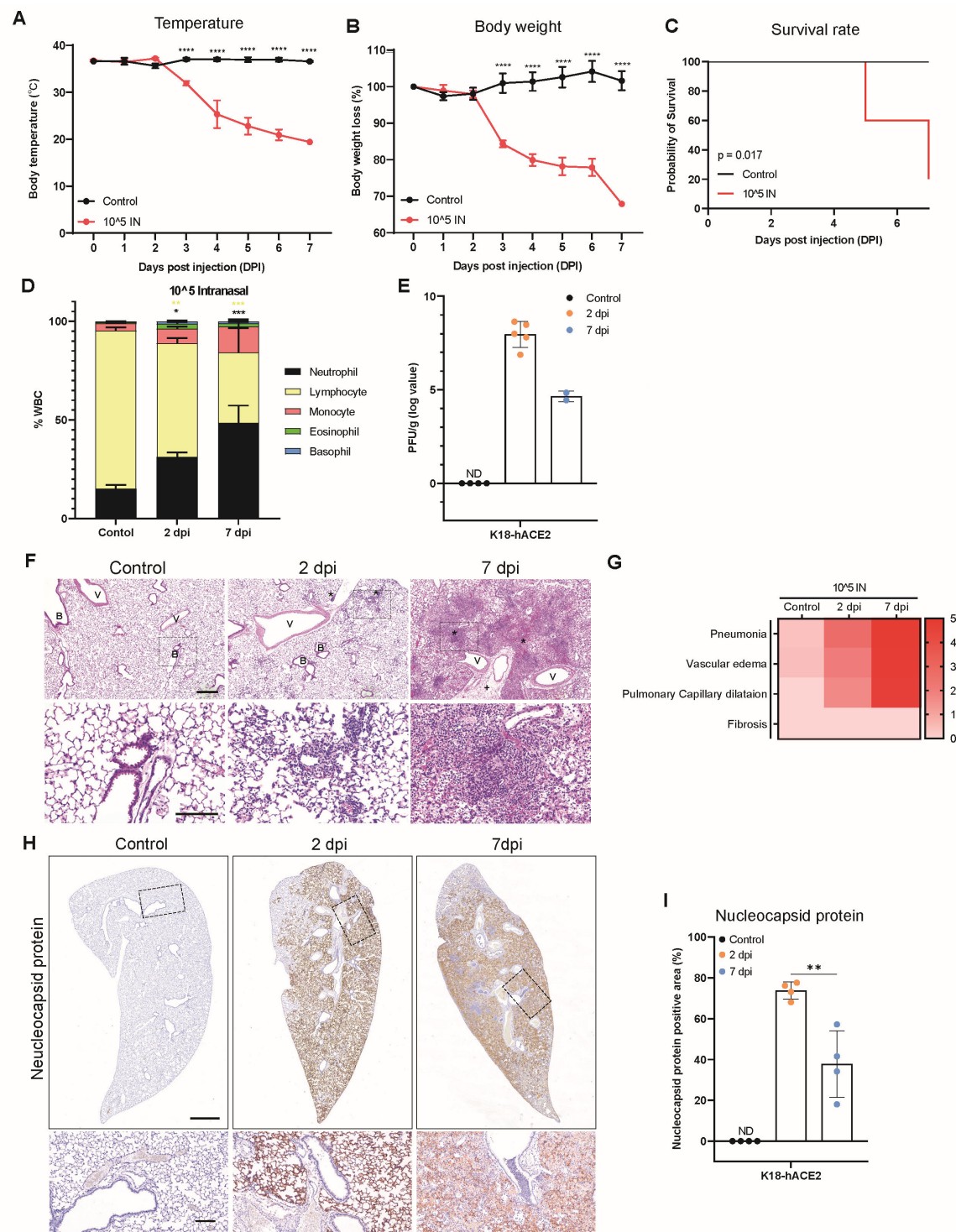

**FIG 1** Clinical features and lung histopathological analysis of SARS-CoV-2-infected K18-hACE mice. (A–C) Preclinical features, body temperatures, body-weight losses, and survival rates of K18-hACE2 mice intranasally infected with $1 \times 10^5$ PFUs of SARS-CoV-2 ($n = 6$) and non-infected control mice ($n = 5$). P-values were calculated using two-way analysis of variance followed by Tukey's multiple comparisons test (A and B) (****$P < 0.001$). Survival rate data were analyzed using the Kaplan–Meier method and compared using the log-rank test (C). (D) White blood cell counts in peripheral blood samples from intranasally infected K18-hACE2 mice and non-infected control mice at each time point. P-values were obtained by performing two-tailed unpaired Student's t-tests (*$P < 0.05$; **$P < 0.01$; ****$P < 0.001$; n.s., not significant). All data are presented as the mean ± s.e.m. (E) Viral loads in the lungs, as determined through plaque assays. (F) Hematoxylin and eosin (H&E) staining of lungs from K18-hACE2 mice after intranasal infection with SARS-CoV-2. Lungs from the infected mice showed evidence of pneumonia (*) and vascular edema (+). Scale bars: 500 µm (top rows); 100 µm (bottom rows). B, bronchiole; V, vessel. (G) Heatmap showing the histopathological scores (Continued on next page)

Fig 1 (Continued)

of lungs from infected K18-hACE2 mice and non-infected (control) mice. The severity of the scores ranged from 0 to 5 (0, none; 1, weak; 2, mild; 3, moderate; 4, severe; 5, markedly severe). (H) Immunohistochemical (IHC) images showing N protein expression in the lungs of K18-hACE2 mice intranasally infected with SARS-CoV-2. Scale bars: 1 cm (top rows); 100 µm (bottom rows). (I) Quantification of the percent N protein-positive areas in mouse lungs. *P*-values were obtained by performing two-tailed unpaired Student's *t*-tests (**$P < 0.01$; n.s., not significant). All data are presented as the mean ± s.e.m.

## K18-hACE2 mice were not infected with SARS-CoV-2 via contact transmission

To investigate the occurrence of SARS-CoV-2 contact transmission in mice (which can occur in humans), K18-hACE2 mice were divided into recipient groups that were co-housed with SARS-CoV-2 intranasally infected donor K18-hACE2 mice at 2 dpi (2DC) or 6 dpi (6DC) for 48 h (Fig. 2A). A 48 h co-housing period was implemented to evaluate SARS-CoV-2 transmissibility at two distinct donor infection stages: an early phase, marked by localized infection in the respiratory tract and oral cavity, and a late phase with systemic viral dissemination. Our aim was to determine whether contact transmission could be initiated during early localized viral shedding vs more advanced systemic infection. First, clinical signs were confirmed; neither the 2DC- nor 6DC-group mice showed significant changes in body temperature or weight (Fig. 2B and C). These mice subjected to contact transmission did not die during the experiment (Fig. 2D). Analysis of the composition of white blood cells in the peripheral blood revealed no changes in monocytes, eosinophils, basophils, or the neutrophil: lymphocyte ratio (Fig. 2E). In addition, neutralizing antibodies were not detected in serum samples from the co-housed mice (Fig. 2F). These data indicate that mice in contact with SARS-CoV-2-infected mice did not show any clinical signs.

Subsequently, we performed histopathological analyses of both the pulmonary and extrapulmonary organs. No histopathological changes were observed in the lungs of mice in the 2DC and 6DC groups (Fig. 3A through C), and PFU measurements with isolated lungs did not detect SARS-CoV-2 (Fig. 3D). Likewise, the N protein was not expressed in the nasal conchae (the initial route of respiratory infections) in the 2DC and 6DC groups at any time point (Fig. 3E through J). To enable quantitative assessment, RT-qPCR was performed on RNA extracted from lung tissues of both donor and contact groups. SARS-CoV-2 RNA targeting the surface glycoprotein gene was detected in donor mice with Ct values below 25, confirming active infection. In contrast, all contact and negative control mice exhibited Ct values of 38 or higher, or undetectable levels, indicating the absence of viral RNA (Fig. S5A).

A pathological analysis of the extrapulmonary organs, including the spleen, small intestine, and brain, was also conducted. Unlike the pathological changes observed in the donor group, no lesions were found in any of the extrapulmonary organs of the recipient group (Fig. S4A through I).

Taken together, our data revealed the absence of clinical and pathological changes in the pulmonary and extrapulmonary organs of K18-hACE2 mice in contact with SARS-CoV-2-infected mice.

## Immune responses did not change in mice subjected to potential SARS-CoV-2 contact transmission

Following SARS-CoV-2 infection, a dynamic immune response occurs in the lungs, including the activation of innate immunity through neutrophils and the subsequent activation of adaptive immunity (51, 52). Therefore, IHC staining was performed to confirm the changes in each immune cell type in the lungs of both the donor and recipient groups. Initially, we focused on key innate immune cells, specifically macrophages and neutrophils. In the donor group, the abundances of F4/80-positive macrophages and Ly6-G/Ly6-C-positive neutrophils increased significantly by 8% and 5%, respectively, when compared with the corresponding abundances in mock-infected mice (Fig. 4A, C, D and F). By 7 dpi, the numbers of macrophages and neutrophils had increased by 22% and 16%, respectively (Fig. 4A, C, D and F). However, the numbers of

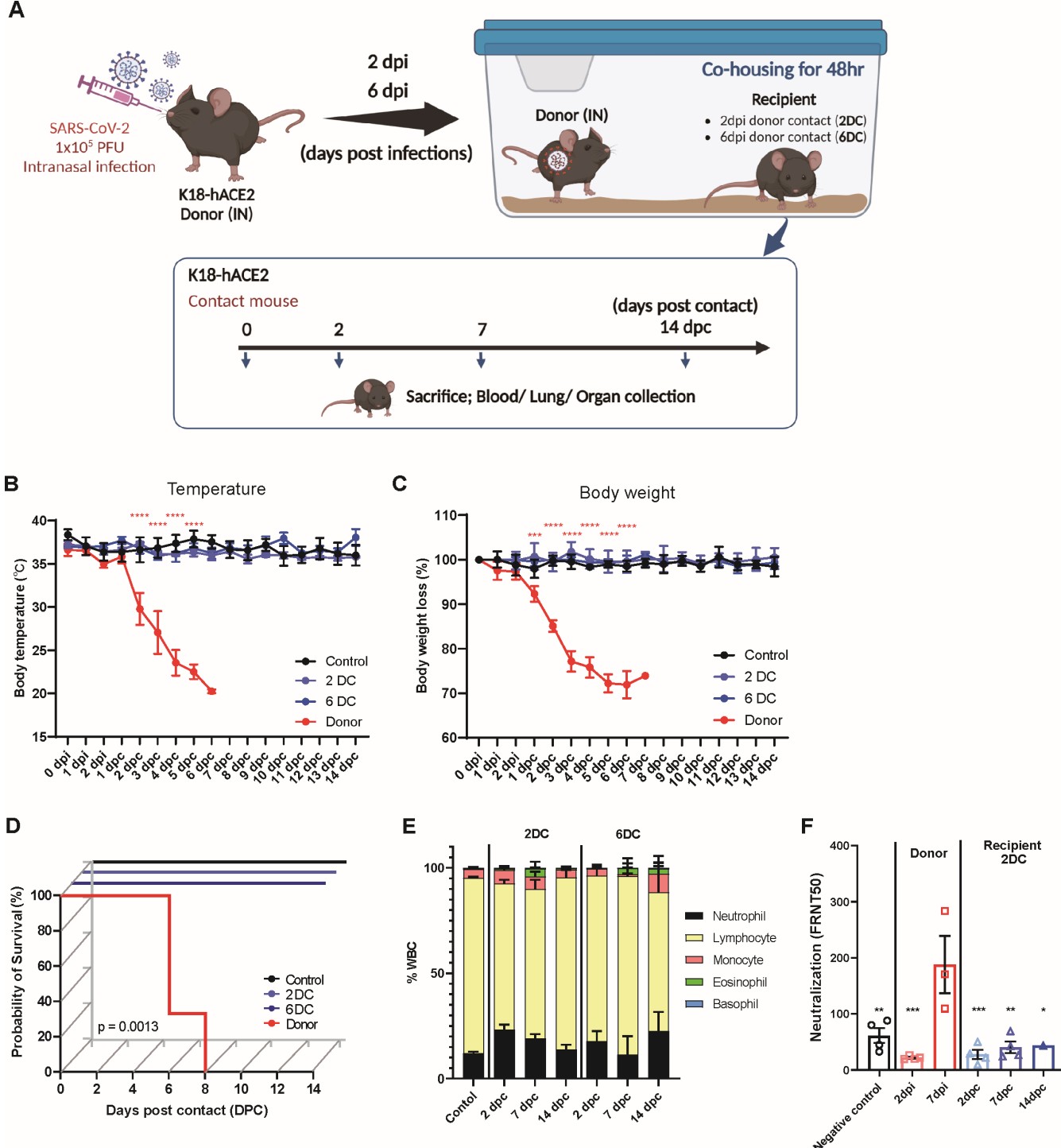

**FIG 2** Mice subjected to contact transmission did not exhibit any changes in clinical features. (A) Schematic image showing a model of contact transmission. (B–D) Preclinical features, body temperatures, body-weight losses, and survival rates of K18-hACE2 mice intranasally infected with $1 \times 10^5$ PFUs of SARS-CoV-2 (donor), recipient mice subjected to donor contact at 2 dpi (2DC group), and mice subjected to donor contact at 6 dpi (6DC), and non-infected control mice ($n = 4$–6/dpi). $P$-values were calculated using two-way analysis of variance followed by Tukey's multiple comparisons test (B and C) (****$P < 0.001$). Survival rate data were analyzed using the Kaplan–Meier method and compared using the log-rank test (D). (E) White blood cell counts in peripheral blood from the 2DC, 6DC, and non-infected control groups at each time point. (F) Neutralizing activity ($FRNT_{50}$) was measured in serum collected from donor mice at 2 and 7 dpi and from recipient 2DC mice at 2, 7, and 14 dpc. $P$-values were obtained by performing two-way analysis of variance followed by Tukey's multiple comparisons test, with comparisons made against the donor 7 dpi group (*$P < 0.05$, **$P < 0.01$, ***$P < 0.005$).

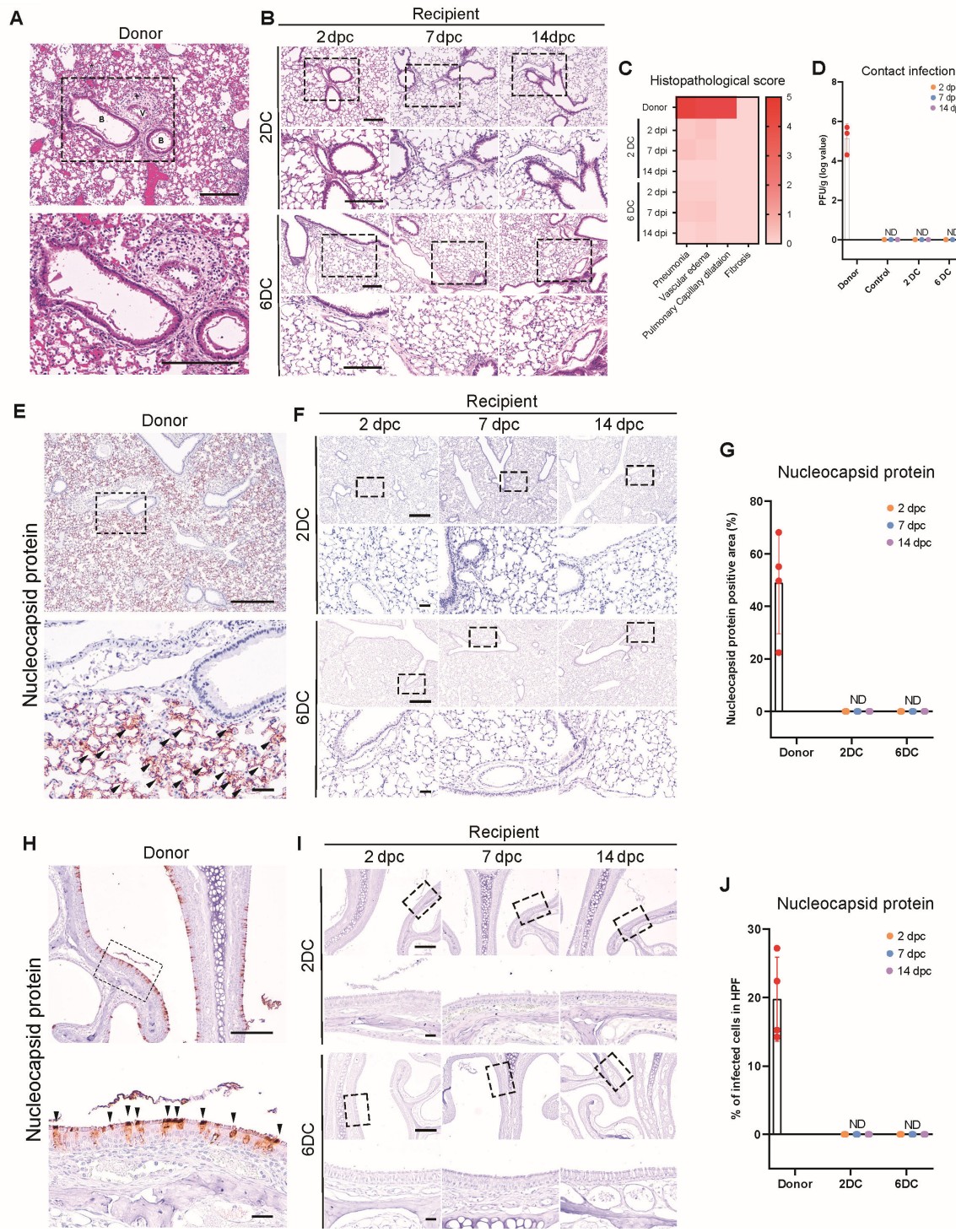

**FIG 3** Respiratory tract infections via contact transmission were not evident in K18-hACE2 mice. (A and B) Hematoxylin and eosin (H&E) staining of lungs from K18-hACE2 donor mice (2 days post-intranasal SARS-CoV-2 infection), recipient mice subjected to donor contact at 2 dpi (2DC group), and mice subjected to donor contact at 6 dpi (6DC group). Infected lungs presented with pneumonia (*) and vascular edema (+). Scale bars: 200 μm. (C) Heatmap showing the histopathological scores of lungs from the donor, 2DC, and 6DC groups. The severity of the scores ranged from 0 to 5 (0, none; 1, weak; 2, mild; 3, moderate; 4, severe; 5, markedly severe). (D) Viral loads in the lungs, based on plaque assays. ND: not detected. (E and F) IHC images of the N protein in mouse lungs from mice in the donor, 2DC, and 6DC groups (*n* = 4–6/dpi). Scale bars: 500 μm (top rows); 100 μm (bottom rows). Black triangles indicate positive cells. (G) Quantification of the percent N protein-positive areas in mouse lungs. ND: not detected. (H and I) IHC images of N protein in mouse nasal conchae from mice in the donor, 2DC, and 6DC groups (*n* = 4–6/dpi). Scale bars: 200 μm (top rows); 50 μm (bottom rows). Black triangles indicate positive cells. (J) Quantification of the percent N protein-positive areas in mouse nasal conchae.

macrophages and neutrophils did not differ significantly between the donor and negative-control group at any day post-contact (Fig. 4B, C, E and F).

In terms of the adaptive immune system, donor mice at 2 dpi showed a 3% increase in CD3b-positive T cells (Fig. 4G and I), whereas the number of PTPRC-positive B cells did not increase (Fig. 4J and L). The numbers of T and B cells were 14% and 6% higher, respectively, in infected mice at 7 dpi than in the negative-control mice (Fig. 4G, I, J and L). In contrast, recipient mice showed no changes in T and B cells in either the 2DC or 6DC group (Fig. 4H, I, K and L), similar to innate immune response data. Our data represent immunological changes that occurred in the lungs of donor and recipient mice. In donor mice, the innate immune system was activated at an early stage, and this activation had increased further at 7 dpi. The adaptive immune system was also activated at 7 dpi, with slightly higher activation found at 2 dpi. In recipient mice, no significant changes were observed in any immune cell population, similar to observations made with control mice. To further evaluate the innate immune response, expression levels of *Cxcl10*, *Ifnb1*, and *Il6* were quantified. These genes showed significant upregulation in donor mice, whereas expression in contact mice remained comparable to uninfected controls (Fig. S6A through C). Overall, considering the pathological data, our findings indicate that contact with SARS-CoV-2-infected K18-hACE2 mice did not lead to SARS-CoV-2 infection.

## Contact infection does not occur even under conditions of immunodeficiency

Previous findings confirmed that infections in immunodeficient mice were more contagious and lasted longer (32, 54). The data led us to hypothesize that the absence of contact infection in previous experiments might have been related to immunity. To test this hypothesis, immunodeficiency was induced in K18-hACE2 mice with an FVB background through drug treatment, and the occurrence of contact transmission was measured (55, 56). The mice were administered MMC to induce immunodeficiency 3 days before contact infection, and a lower MMC concentration was administered following the contact period (Fig. 5A). Fewer white blood cells were observed in the peripheral blood of the MMC-treated mice (Fig. S7A). However, after contact infection, the body weights or temperatures were not different from those in the control group (Fig. S7B and C). Similar to C57BL/6-background K18-hACE2 mice, K18-hACE2 mice with an FVB background died 4 dpi following intranasal SARS-CoV-2 infection, with only one mouse surviving by 6 dpi. However, none of the mice, whether MMC-treated or not, died following contact infection (Fig. S7D). Neutralizing antibodies were not detected in the serum of the MMC-treated recipient group. In contrast to the FVB-background K18-hACE2 donor mice (which showed detectable viral PFUs in their lung tissues), no viral PFUs and SARS-CoV-2 RNA targeting the surface glycoprotein gene were detected in the MMC-treated recipient group (Fig. 5B and C; Fig. S5B). Pathological analysis of the lungs revealed no lesions in the group treated with MMC alone (Fig. S7E). Additionally, the donor group exhibited severe pneumonia, vascular edema, and pulmonary capillary dilatation, similar to C57BL/6-background K18-hACE2 donor mice. However, in the recipient groups, no lung lesions were observed at 2- or 7-days post-contact (dpc) in either the MMC-treated or untreated groups. (Fig. 5D and E). SARS-CoV-2 was not detected in the lungs of the MMC-treated recipient group at 2 and 7 dpc using *in situ* hybridization staining for the *S* gene (Fig. 5F; Fig. S7F). Similarly, the N protein was not detected in the nasal concha, the initial respiratory route (Fig. S7G). The expression levels of *Cxcl10*, *Ifnb1*, and *Il6* in lung tissues did not differ between the MMC-treated recipient group and uninfected controls (Fig. S6D through F). These data suggest that contact infection was not induced, even under conditions of heightened susceptibility to infection due to immunodeficiency.

## DISCUSSION

In humans, SARS-CoV-2 infection is caused by airborne transmission or direct contact with an infected individual (14, 41), and animal experiments are being conducted to model these infection routes (18). Among animals with confirmed contact infections,

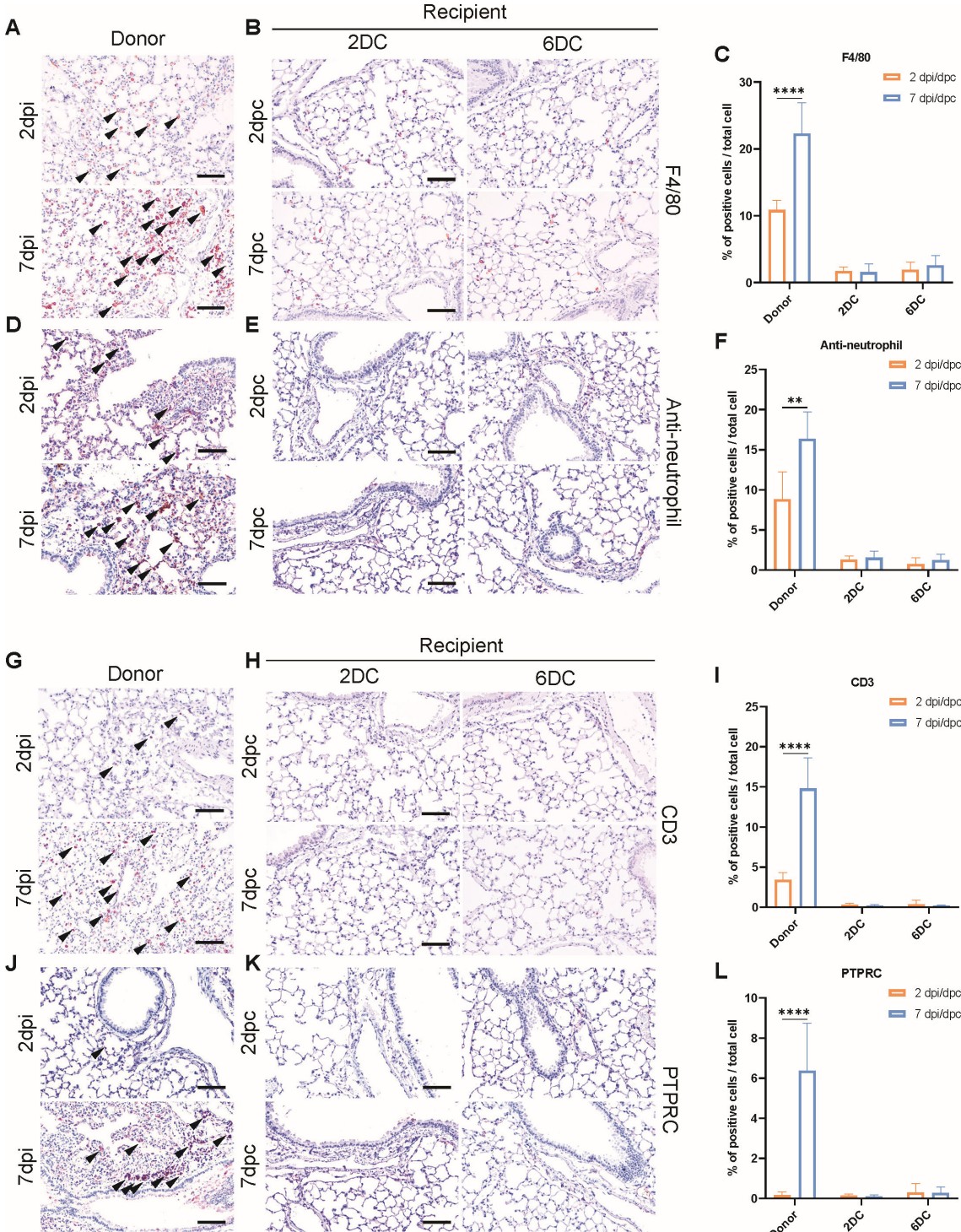

**FIG 4** Comparison of immune-cell distributions between SARS-CoV-2-infected mice and mice subjected to contact transmission to confirm immune responses following infection. IHC images are shown for immune cells from the donor mice, recipient mice subjected to donor contact at 2 dpi (2DC group), and mice subjected to donor contact at 6 dpi (6DC group). (A and B) F4/80 (macrophage marker), (D and E) Ly-6G/Ly-6C (neutrophil marker) (53), CD3B (T cell marker), and (J and K) PTPRC (B cell marker) detection. Scale bars: 100 µm (bottom row). (C, F, I, and L) Quantification of the percentages of F4/80-positive (C), Ly-6G/Ly-6C-positive (F), CD3B-positive (I), and PTPRC-positive (L) cells among total cells. *P*-values were determined by performing two-tailed unpaired Student's *t*-tests (**\**P* < 0.01; \*\*\*\**P* < 0.001; n.s., not significant). All data are presented as the mean ± s.e.m. Black triangles indicate positive cells (A, D, G, and H).

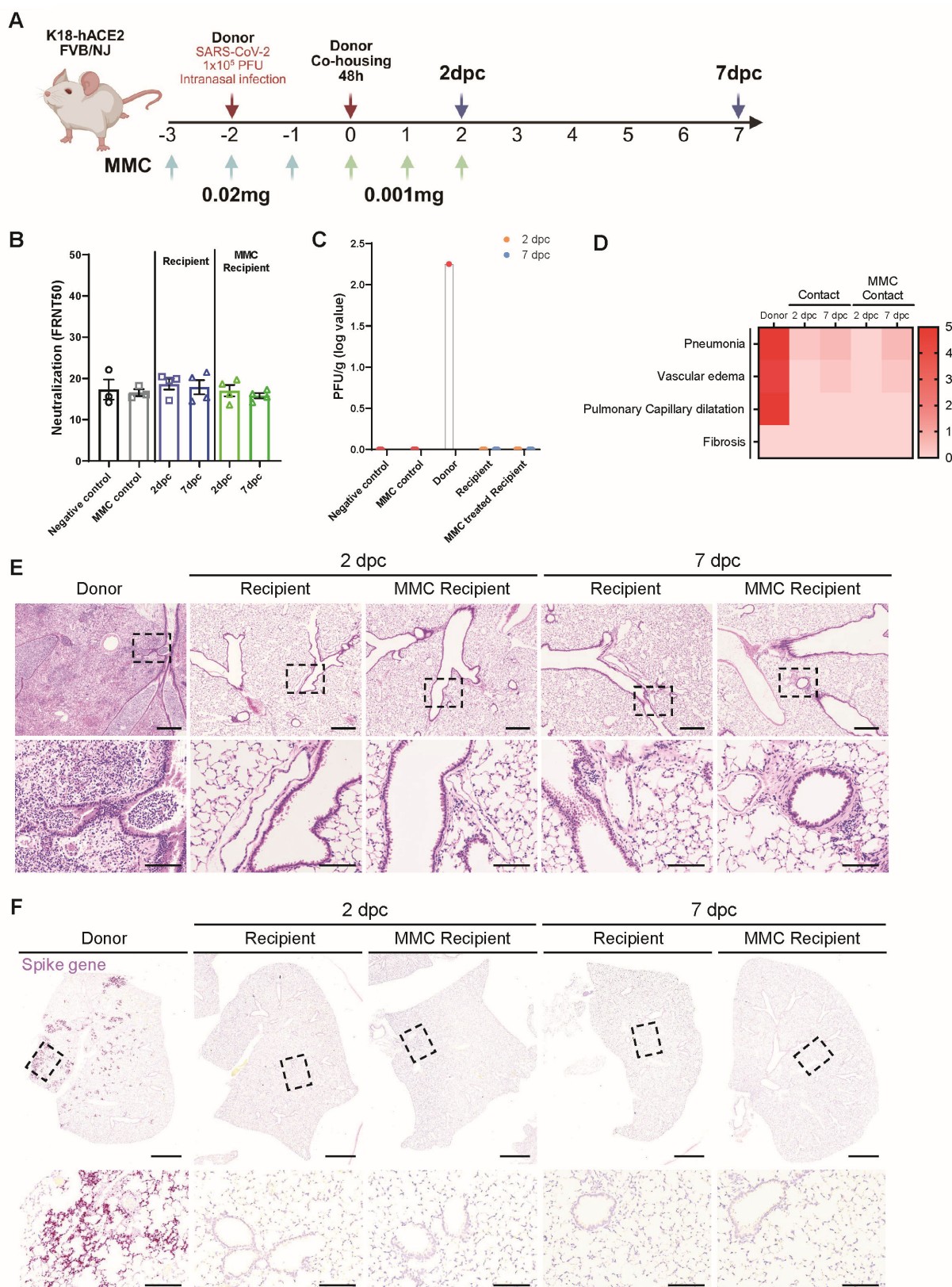

**FIG 5** Signs of transmission infection did not appear in FVB-background K18-hACE2 mice, regardless of their immunodeficiency status. (A) Schematic image showing a model of contact transmission and induced immunodeficiency through MMC treatment. Donor and recipient mice were co-housed for 48 h before assessing transmission outcomes. (B) Neutralizing-antibody levels in recipient and MMC-treated recipient mice. (C) Viral loads in the lungs, based on plaque (Continued on next page)

**Fig 5 (Continued)**

assays. (D) Heatmap showing histopathological scores for lungs from donor, recipient, and MMC-treated recipient mice. The severity of the scores ranged from 0 to 5 (0, none; 1, weak; 2, mild; 3, moderate; 4, severe; 5, markedly severe). (E) Hematoxylin and eosin (H&E) staining of lungs from FVB-background donor K18-hACE2 mice (2 days post-intranasal SARS-CoV-2 infection), recipient mice, and MMC-treated recipient mice subjected to donor contact for 2 days when the donors were at 2 and 7 dpc. Scale bars: 500 µm (top rows); 100 µm (bottom rows). (F) Representative *in situ* hybridization images of the *S* gene in the lungs of donor, recipient, or MMC-treated recipient mice at 2 dpc and 7 dpc. Scale bars: 1 mm (top); 200 µm (bottom).

ferrets, cats, and hamsters can be infected through direct contact, whereas mice exhibit various responses to contact infection after SARS-CoV-2 infection. In a study by Rodriguez-Rodriguez et al., transmission experiments were conducted using adult K18-hACE2 mice and the WA-1 variant (A.1 lineage) (43). They reported only one case of contact infection. However, no decrease in body weight was observed in the control mice. Similarly, when mice were infected with the beta variant (B.1.351; which can cause contact infection), no decrease in body weight occurred, and the virus was detected in the trachea but not in the lungs (37). These data indicate the uncertainty regarding contact infections in mouse models.

In this study, we assessed the effectiveness of contact infection in K18-hACE2 mice under various conditions. Before inducing contact infection, we intranasally infected K18-hACE2 mice with SARS-CoV-2 to observe their overall responses to the virus. The responses were similar to those of previous studies, confirming that these mice could serve as effective donors for contact-infection experiments (35, 45).

We used the results from 2 dpi, where high viral titers were observed in the respiratory organs, but no severe lesions appeared in other organs or the lungs. We also used the results from 7 dpi, when the mice exhibited weight loss, reduced body temperature, and death, showing severe lesions in both respiratory and non-respiratory organs. Based on these observations, we hypothesized that contact during the early stages of infection might differ from contact during the later stages. To test this hypothesis, we induced contact infection in both environments. However, under both donor conditions, no symptoms of infection were observed in the co-housed mice. These data suggest that contact infection was not induced in K18-hACE2 mice, regardless of the timing of the infection.

We also induced contact infection using FVB-background K18-hACE2 mice in addition to C57BL/6-background mice. Previous reports have indicated that FVB-background mice exhibit varying levels of cytokine secretion in the context of infection or disease development compared to C57BL/6-background mice. This difference in cytokine response can lead to variations in susceptibility to diseases between these mouse strains (45, 57–59).

Finally, given prior research suggesting that the susceptibility to SARS-CoV-2 infection can increase under immunodeficient conditions, we induced immunodeficiency by administering mice MMC intraperitoneally (32). However, contact infection still did not occur after inducing immunodeficiency. These findings suggest that K18-hACE2 transgenic mice are insufficient as a model for studying transmission through contact infection, regardless of their background or immune status.

We attempted to induce contact infection in K18-hACE2 mice under various conditions, but no signs of infection were observed, strengthening the credibility of the claim that contact infection does not occur in K18-hACE2 mice. However, a limitation of our study is that we did not identify the specific transmission pathways that were impaired to explain why contact infection failed in K18-hACE2 mice. We focused solely on assessing the contact transmissibility of the Beta variant of SARS-CoV-2, without evaluating the contact infection potentials of other variants. Previous reports indicate that responses to infection can vary depending on the viral strain, and some evidence suggests that transmission may not occur with certain strains (37, 39, 44). However, considering the continuously evolving nature of SARS-CoV-2, the possibility of contact transmission in mice with other emerging variants should not be ruled out. Further studies are necessary to explore this potential.

Previous reports have demonstrated the presence of detectable virus in the oral cavities or feces of SARS-CoV-2-infected mice. In this study, we did not measure SARS-CoV-2 titers in the nasal or oral mucosal areas of donor and recipient mice during contact infection. However, we confirmed that viral RNAs and proteins were expressed in the nasal concha following infection through *in situ* analysis and immunostaining. Additionally, clinical signs, serum neutralizing-antibody levels, and pathological analysis all suggested that minimal contact transmission occurred. Although previous research has shown that airborne SARS-CoV-2 transmission can lead to infection in K18-hACE2 mice, limited data exist on the viral concentrations necessary to induce infection through contact transmission (53). Also, our study used the K18-hACE2 mouse model, in which hACE2 expression is driven by the human keratin-18 promoter. This promoter induces robust expression in the lower respiratory tract, particularly in alveolar epithelial cells, but only limited expression in the upper airway epithelium, including the nasal and bronchial regions. Consequently, SARS-CoV-2 infection in K18-hACE2 mice primarily targets the lungs, with minimal replication in the upper airways. Supporting this, previous studies have shown that by 3 dpi, viral RNA levels in the lungs of K18-hACE2 mice can reach approximately $10^{10}$ copies/g, whereas levels in the nasal epithelium are up to $10^5$-fold lower (46). Similarly, Winkler et al. reported that infectious viral titers in the nasal turbinate were approximately 100-fold lower than those in the lungs at 2 dpi (52). These findings indicate that although SARS-CoV-2 can reach the upper airways in K18-hACE2 mice, the replication levels there are insufficient to sustain efficient contact transmission. The rapid disease progression and high mortality in K18-hACE2 mice may further limit the window of viral shedding, reducing transmission opportunities. In such highly susceptible hosts, the disease course is skewed toward severe lower respiratory tract involvement rather than prolonged upper respiratory tract infection, which is typically necessary for efficient spread.

In contrast, the transgenic hACE2 mouse model used by Bao et al. was generated using the endogenous mouse ACE2 promoter, which drives hACE2 expression in a pattern more consistent with native ACE2 distribution, including robust expression in the nasal and airway epithelium (41). This difference in tissue tropism likely facilitated higher viral replication in the upper respiratory tract, prolonged viral shedding, and consequently, successful contact transmission.

Taken together, our findings support the use of adult K18-hACE2 mice for studying intranasal SARS-CoV-2 infection, while also highlighting their limitations as a model for direct-contact transmission. This limitation likely reflects differences in hACE2 transgene regulation and expression patterns compared with other hACE2 models, which can profoundly influence both disease pathogenesis and transmission potential. Improved mouse models are needed to better study alternative transmission routes, such as contact transmission. Further research is required to explore the underlying factors contributing to the resistance of K18-hACE2 mice to contact infection. These insights are critical for refining animal models and deepening the understanding of SARS-CoV-2 transmission, which will ultimately aid in developing more effective prevention and control measures during the ongoing pandemic.

## MATERIALS AND METHODS

### Animals

K18-hACE2 mice with a C57BL/6 background were purchased from Jackson Laboratory (Bar Harbor, ME, USA), and K18-hACE2 mice with an FVB/NJ background were produced and provided by the Korea Mouse Phenotyping Center at Seoul National University. All animal use protocols were approved by the Institutional Animal Care and Use Committee (2020-0216 and BA-2008-301-071-03) of the Yonsei University College of Medicine and were accredited by the Association for Assessment and Accreditation of Laboratory Animal Care International (#001071). All procedures were conducted in a biosafety level

3 facility in accordance with the Public Health Service Policy on Human Care and Use of Laboratory Animals.

## Virus

For virus production, we obtained the Vero African green monkey kidney cell line (KCLB 10081) from the Korean Cell Line Bank and cultured it in Dulbecco's minimum Eagle's medium supplemented with 2 mM L-glutamine, 100 units/mL penicillin, 100 µg/mL streptomycin, and 5% fetal bovine serum. The cells were maintained at 37°C in a humidified incubator with 5% $CO_2$. SARS-CoV-2 was obtained from the National Culture Collection for Pathogens of Osong, Korea (NCCP 43326, S-type variant). Virus titers were measured through plaque assays, as previously described (60). Briefly, supernatants from infected cells or homogenates from infected tissue samples were serially diluted. The supernatants were added to Vero cells that had been seeded in a six-well plate and incubated at 37°C for 1 h with gentle agitation every 15 min. The cells were then overlaid with Dulbecco's minimum Eagle's medium, 1% SeaPlaque agarose (Lonza), 2% fetal bovine serum, 100 units/mL penicillin, and 100 mg/mL streptomycin. After 3 days of incubation, the overlays were removed, and the cells were fixed with 4% paraformaldehyde and stained with a 0.5% crystal violet in a 20% methanol solution. The plaques were counted and multiplied by the dilution factor to quantify the viral titers.

## Contact-transmission model

To study SARS-CoV-2 contact transmission in mice, 12 9-week-old male K18-hACE2 mice were placed in six cages, with two mice in each cage. The mice were anesthetized with a Zoletil–Rompun mixture (4:1) and intranasally inoculated with $1 \times 10^5$ PFUs of SARS-CoV-2 to generate a donor group. In the recipient group, uninfected K18-hACE2 mice were co-housed with donor mice (2:4 ratio) for 48 h and then separated from the donor mice. The mice were divided into the following three groups (12 mice/group): a control group and recipient mice, co-housed with infected intranasally donor mice at 2 dpi (2DC group) or 6 dpi (6DC group). For the mock (control) infection, the donor group was administered an equal volume of phosphate-buffered saline (PBS). Mouse body weights and temperatures were monitored daily using an electronic scale and an implantable programmable temperature transponder (IP55-300; Bio Medic Data Systems, USA). The mice were euthanized 2, 7, and 14 dpc for further analysis (4 mice/time point).

## MMC treatment

Immunosuppression was induced in FVB/NJ-background K18-hACE2 mice via intraperitoneal daily injections of MMC (0.02 mg/mouse), starting 3 days before infection, followed by 0.001 mg/mouse for 3 days after infection.

## Histopathological analysis

For histopathological analysis, we fixed lung, spleen, small intestine, and brain tissues in 10% neutral-buffered formalin (F5554, Sigma, St. Louis, MO, USA) for 24 h and embedded them in paraffin. The fixed samples were sliced into 4 µm-thick tissue sections for hematoxylin and eosin (H&E) and IHC staining using a microtome (Leica, Wetzlar, Germany). For H&E staining, sections were de-paraffinized through immersion three times in xylene and rehydrated sequentially in 100%, 95%, and 70% ethanol. The slides were stained with 0.1% Mayer's hematoxylin (3309, Agilent, Santa Clara, CA, USA) for 10 min and then dipped into a 0.5% Eosin Y (F5554, Sigma) solution. The slides were then washed in distilled water until the eosin stopped forming streaks and dehydrated in an ascending serial gradient of ethanol (50%, 70%, 95%, and 100%) for 1 min/dehydration step. The slides were covered with a mounting solution (6769007, Thermo Scientific, Waltham, MA, USA) and analyzed under a light microscope (BX43, Olympus, Tokyo, Japan). Histopathological analyses were performed by an animal pathologist (K.T.N.).

Histopathological severity was scored on a scale from 0 to 5 (0, none; 1, weak; 2, mild; 3, moderate; 4, severe; 5, markedly severe).

## IHC staining

For IHC staining, paraffin-embedded samples were sliced into 4 µm-thick sections. The sections were de-paraffinized via immersion thrice in xylene, followed by rehydration using a descending graded series of ethanol. Using pH 6.0 antigen-retrieval solution (S1699, Agilent), antigens were retrieved by pressing and boiling the sections in a high-pressure cooker for 15 min. Subsequently, the sections were cooled on ice for 1 h, washed twice with Dulbecco's PBS, and incubated in 3% $H_2O_2$ for 30 min to block endogenous peroxidase activity. The sections were then washed twice with PBS and incubated in a protein-blocking solution (X0909, Agilent) for 2 h at 22°C in a humidified chamber. When using mouse or rat primary antibodies, the M.O.M. Kit (BMK-2202, Vector Laboratories, Newark, CA, USA) was employed before protein blocking. The slides were incubated overnight at 4°C with primary antibodies against the SARS-CoV-2 N protein (40143-MM08, Sino Biological, 1:1,000), F4/80 (70076, Cell Signaling Technology, 1:1,000), CD3b (ab5690, Abcam, 1:1,000), anti-PTPRC (ab64100, Abcam, 1:1,000), and neutrophils (ab2557, Abcam, 1:2,000). The sections were then incubated with a horseradish peroxidase-conjugated secondary antibody (K4001, Agilent) for 15 min or with biotinylated anti-rat IgG (Vector) for 30 min, followed by incubation with the ABC reagent (Vector) for 30 min at room temperature. To develop the horseradish peroxidase-labeled antibodies on the sections, 3,3′-diaminobenzidine in chromogen solution (K3468, Agilent) was diluted in 20 µL to 1 mL of imidazole-HCl buffer, containing hydrogen peroxide (K3468, Agilent) and applied to the sections for 15–30 seconds to detect the signals. Mayer's hematoxylin (3309, Agilent) was used to counterstain the nuclei. After counterstaining, washing and dehydration steps were performed, and the slides were covered with mounting solution (6769007, Thermo Scientific).

## In situ hybridization

For in situ hybridization staining, the human ACE2 RNA probe and RNAscope 2.5 HD Red Assay were purchased from Advanced Cell Diagnostics ([ACD], Bio-Techne, MN, USA). In situ hybridization was performed according to the manufacturer's instructions. Briefly, paraffin-embedded sections were de-paraffinized in xylene and dehydrated twice with 100% ethanol. After air-drying, the slides were treated with $H_2O_2$, boiled in buffer to retrieve the targets, and treated with protease K for 30 min. The sections were incubated for 2 h with the RNA probe, after which the RNA signals were amplified using the amplifying reagent (322360, ACD) and detected with the Fast Red reagent (322360, ACD).

## Hematological analysis

For hematological analysis, peripheral blood samples were collected from mouse hearts following euthanasia using a 1 mL syringe. The collected blood samples were transferred into 1.5 mL microtubes containing 20 µL 0.5 M ethylenediaminetetraacetic acid to prevent blood clotting. Complete blood counts were performed using a hematology analyzer (BC-5000, Mindray Global, Nanshan, China).

## Statistical analysis

Statistical analyses were performed using the GraphPad Prism software (v9.0). Statistical significance was determined using one-way analysis of variance with Šidák's multiple-comparison test. All data are presented as the mean ± SD. A value of $P < 0.05$ was considered to represent a statistically significant difference.

## ACKNOWLEDGMENTS

The authors declare that the research was conducted in the absence of any commercial or financial relationships that could be construed as a potential conflict of interest.

## AUTHOR AFFILIATIONS

[1]Department of Biomedical Sciences, Graduate School of Medical Science, Brain Korea 21 Project, Yonsei University College of Medicine, Seoul, South Korea

[2]Department of Molecular and Life Science, Hanyang University, Ansan, Republic of Korea

[3]Department of Clinical Drug Discovery & Development, Severance Biomedical Science Institute, College of Medicine, Yonsei University, Seoul, Republic of Korea

[4]Institute of Immunology and Immunological Diseases, Yonsei University College of Medicine, Seoul, South Korea

[5]Department of Microbiology, Yonsei University College of Medicine, Seoul, South Korea

[6]Korea Mouse Phenotyping Center (KMPC), Seoul National University, Seoul, South Korea

[7]Laboratory of Developmental Biology and Genomics, Research Institute for Veterinary Science, and BK 21 PLUS Program for Creative Veterinary Science Research, College of Veterinary Medicine, Seoul National University, Seoul, South Korea

[8]BIO MAX Institute, Seoul National University, Seoul, South Korea

[9]Interdisciplinary Program for Bioinformatics, Seoul National University, Seoul, South Korea

## AUTHOR ORCIDs

Jiseon Kim http://orcid.org/0000-0002-0979-0278
Sung-Hee Kim http://orcid.org/0000-0002-2970-9449
Donghun Jeon http://orcid.org/0000-0002-9161-6198
Haengdueng Jeong http://orcid.org/0000-0002-9218-7372
Jun-Young Seo http://orcid.org/0000-0003-4004-2013
Je Kyung Seong http://orcid.org/0000-0003-1177-6958
Ki Taek Nam http://orcid.org/0000-0001-5292-1280

## FUNDING

| Funder | Grant(s) | Author(s) |
|---|---|---|
| Bio & Medical Technology Development Program of the National Research Foundation (NRF) funded by the Korean government (MSIT) | RS-2021-NR057630, RS-2024-00400118, RS-2022-NR070588, RS-2022-NR067350, RS-2024-00443043 | Ki Taek Nam |
| Nano & Material Technology Development Program through the National Research Foundation of Korea (NRF) funded by Ministry of Science and ICT | RS-2024-00407093 | Ki Taek Nam |
| Hur Jiyoung Foundation | | Jiseon Kim |

## AUTHOR CONTRIBUTIONS

Jiseon Kim, Conceptualization, Data curation, Formal analysis, Investigation, Project administration, Resources, Supervision, Visualization, Writing – original draft | Sung-Hee Kim, Conceptualization, Data curation, Formal analysis, Investigation, Project administration, Resources, Supervision, Visualization, Writing – original draft | Donghun Jeon, Conceptualization, Data curation, Formal analysis, Validation, Visualization, Writing – original draft | Haengdueng Jeong, Conceptualization, Data curation, Supervision, Validation, Visualization, Writing – original draft | Chanyang Uhm, Formal analysis, Investigation, Writing – review and editing | Heeju Oh, Formal analysis, Investigation, Writing – original draft | Kyungrae Cho, Data curation, Formal analysis, Investigation,

Writing – original draft | Yejin Cho, Conceptualization, Data curation, Investigation, Supervision, Validation, Writing – review and editing | Sumin Hur, Conceptualization, Supervision, Validation, Writing – original draft | In Ho Park, Conceptualization, Data curation, Investigation, Resources, Visualization, Writing – original draft | Jooyeon Oh, Data curation, Investigation, Resources, Visualization, Writing – original draft | Jeong Jin Kim, Data curation, Formal analysis, Resources, Validation, Writing – original draft | Jun-Young Seo, Data curation, Resources, Supervision, Writing – review and editing | Jeon-Soo Shin, Data curation, Resources, Supervision, Writing – review and editing | Je Kyung Seong, Data curation, Supervision, Writing – review and editing | Ki Taek Nam, Conceptualization, Data curation, Supervision, Writing – review and editing

## DATA AVAILABILITY

Data sharing is not applicable for this article because no data sets were generated or analyzed in the current study.

## ADDITIONAL FILES

The following material is available online.

### Supplemental Material

**Supplemental figures (Spectrum03413-24-s0001.pdf).** Figures S1 to S7.

### Open Peer Review

**PEER REVIEW HISTORY (review-history.pdf).** An accounting of the reviewer comments and feedback.

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
