## [Reviewer comments · Microbiology Spectrum]

Microbiology Spectrum

Adult K18-hACE2 Mice Are Suitable for Studying Intranasal SARS-CoV-2 Infection but Not Direct-Contact Transmission

Jiseon Kim, Sung-Hee Kim, Donghun Jeon, Haengdueng Jeong, Chanyang Uhm, Heeju Oh, Kyungrae Cho, Yejin Cho, Sumin Hur, In Ho Park, Jooyeon Oh, Jeong Jin Kim, Jun-Young Seo, Je Kyung Seong, and Ki Taek Nam

Corresponding Author(s): Ki Taek Nam, Yonsei University Brain Korea 21 Project for Medical Science

Review Timeline:

Submission Date:	December 28, 2024
Editorial Decision:	April 4, 2025
Revision Received:	June 9, 2025
Editorial Decision:	July 28, 2025
Revision Received:	August 5, 2025
Accepted:	September 5, 2025

Editor: Luciana Costa

Reviewer(s): The reviewers have opted to remain anonymous.

Transaction Report:

DOI: <https://doi.org/10.1128/spectrum.03413-24>

Re: Spectrum03413-24 (Adult K18-hACE2 Mice Are Suitable for Studying Intranasal SARS-CoV-2 Infection but Not Direct-Contact Transmission)

Dear Prof. Ki Taek Nam:

Thank you for the privilege of reviewing your work. Below you will find my comments, instructions from the Spectrum editorial office, and the reviewer comments.

Please, respond to all the points and concerns raised by both reviewers.

Revision Guidelines

Sincerely,
Luciana Costa
Editor
Microbiology Spectrum

Reviewer #1 (Comments for the Author):

The manuscript entitled "Adult K18-hACE2 Mice Are Suitable for Studying Intranasal SARS-CoV-2 Infection but Not Direct-Contact Transmission" systematically analyzed the model for its use for studying contact transmission. The observation in the animal clinical scores and histological analyses for immune marker staining as well as virus antigen staining and infectious virus production clearly demonstrates that the contact transmission is not occurring

Strength: The manuscript is well written with all details down for the executed experiments. The research is reconfirming earlier observations that transmission of SARS-CoV-2 is not efficient between mice through robust histopathological as well as other assays.

Weakness: Given that the respiratory droplet transmission rate is not known for the mice under study, the study could have been more interesting if some other biological elements were added to it. For eg: whether the donor mother can transmit to the neonates or infants. Including the treatment of MMC to demonstrate the immunodeficiency mice as recipients is good. However, the read out of MMC treatment is missing in the sense, a positive control for the treatment of MMC is missing.

Major concern: As mentioned, the article studies the contact transmission and their statement on the conclusion " K18-hACE2 transgenic mice should be considered an optimal model for intranasal infection with SARS-CoV-2," needs to be restated as we cannot consider it as optimal and it depends on type of research undergoing. Hence, I think the title is a bit appropriate.

Minor concerns

Figure 1

(i) Stats are missing from Fig 1A-C, or if not significant mention it.

(ii) Mention the histological scoring for each pathological change in methodology

Figure 2

(i) Provide Stats for Fig 2B-D, or if not significant mention it.

Figure 3,4

(ii) It would be great if you can mark the positive cells of any staining in the donor tissue section for the ease of understanding the histology.

Reviewer #2 (Comments for the Author):

This study investigates the contact transmission of the SARS-CoV-2 Beta variant in K18-hACE2 mice. Given that K18-hACE2 mice are widely used to study SARS-CoV-2 biology and pathogenesis, understanding the mode of transmission in this commonly used infection model is crucial. However, several clarifications are needed to strengthen the manuscript.

Major points:

- 1) Line 152: The authors state that the viral load was the highest at 2 days post-infection (dpi) and declined by 7 dpi. However, were viral loads measured at other time points? It is possible that the peak occurred on day 3, which should be clarified.
- 2) Figure 2: Why was the co-housing experiment conducted for only 48 hours? A rationale for this timeframe should be provided.
- 3) The study uses a high infection dose of 10^5 PFU of SARS-CoV-2 in K18-hACE2 mice. What is the justification for administering such a high viral dose?
- 4) Figure 3D: Was the viral titer measured from a donor at 2 dpi or 6 dpi? If it was 2 dpi, the viral loads appear much lower than in Figure 1E. Please clarify.
- 5) Viral RNA levels should be measured using RT-qPCR in the co-housing experiments for more quantitative analysis. Innate immune responses can also be assessed using RT-qPCR, as it provides a more sensitive and quantitative measure.
- 6) Figure 5: The viral titer in donor lung tissue (Figure 5C) is unexpectedly low, despite infection with 10^5 PFU/ml of the virus. This raises concerns: if the FVB background K18-hACE2 mice are not efficiently infected, then the interpretation of the contact transmission experiment may be inherently limited.
- 7) Figure S4B and S4D do not include donor body weight and temperature data. These should be provided.
- 8) Why was the co-housing duration reduced from 48 hours to 24 hours in the MMC-treated experiments? Please clarify the reasoning for this modification.
- 9) Statistical tests are missing for some figures.

Minor points:

- 1) Please clearly mention the variant of SARS-CoV-2 instead of the clade. The authors are using it interchangeably which is confusing for the readers.
- 2) A 2020 (Bao et. al. JID) study reported contradictory results on the contact transmission of SARS-CoV-2 in K18-hACE2 mice. The authors should discuss this prior study in more detail and explain why their findings are different.

Reviewer #1 (Comments for the Author):

Major concern: As mentioned, the article studies the contact transmission and their statement on the conclusion " K18-hACE2 transgenic mice should be considered an optimal model for intranasal infection with SARS-CoV-2," needs to restated as we cannot consider as optimal and it depends on type of research undergoing. Hence, I think the title is a bit appropriate.

Response: We thank the reviewer for this suggestion. We have revised the conclusion to avoid using the term "optimal," which may overstate the applicability of the K18-hACE2 model across different research of SARS-CoV-2. Instead, we have rephrased the statement to emphasize the specific suitability of the model for intranasal infection studies while acknowledging its limitations in modeling contact transmission. The edited statement is highlighted in yellow in the revised manuscript (page 15, lines 305–307).

Minor concerns

Figure 1

(i) Stats are missing from Fig 1A-C, or if not significant mention it.

Response: We thank the reviewer for this valuable comment. As per your suggestion, we have quantified the data and included statistical analyses (**Revision Figure 1A-C**, new **Figure 1A-C in the manuscript**). In the 10^5 intranasal group, both body weight and body temperature showed statistically significant differences starting from 3 dpi. Additionally, survival rates were significantly different between groups, and this statistical significance is now clearly depicted in the revised Figures. The manuscript has been revised accordingly, with the edited statements highlighted in yellow (**page 32, lines 704-706**).

Revision Figure 1

Revision Figure 1. (A–C) Preclinical features, body temperatures, body-weight losses, and survival rates of K18-hACE2 mice intranasally infected with 1×10^5 PFUs of SARS-CoV-2 (n = 6) and non-infected control mice (n = 5). P-values were calculated using two-way ANOVA followed by Tukey's multiple comparisons test (A and B) (**P < 0.001). Survival rate data were analyzed using the Kaplan–Meier method and compared using the log-rank test.**

(ii) Mention the histological scoring for each pathological changes in methodology

Response: We thank the reviewer for this suggestion. As recommended, we have included a description of the histological scoring criteria corresponding to each pathological change in the Methods section. This newly added information is highlighted in yellow in the revised manuscript (page 18, lines 374-375).

Figure 2

(i) Provide Stats are Fig 2B-D, or if not significant mention it.

Response: We thank the reviewer for this observation. As shown in Fig. 2B–D, we assessed body weight and body temperature across four groups (Control, Donor, 2DC, and 6DC). No statistically significant differences were observed in the Control, 2DC, and 6DC groups following contact exposure. In contrast, the Donor group exhibited significant decreases in both body weight and temperature due to SARS-CoV-2 infection (**Revision Figure 2A and B**, new **Figure 2B-C in the manuscript**) Regarding survival, only the Donor group exhibited a significant reduction in survival rate, whereas all mice in the contact exposure groups survived (**Revision Figure 2C**, new **Figure 2D in the manuscript**) We have clarified these findings in the figure and updated the legend accordingly, highlighted in yellow in the revised manuscript (page 33, lines 728–730).

Revision Figure 2

Revision Figure 2. (A–C) Preclinical features, body temperatures, body-weight losses, and survival rates of C57BL/6 background K18-hACE2 mice intranasally infected with 1×10^5 PFUs of SARS-CoV-2 (Donor), recipient mice subjected to donor contact at 2 dpi (2DC group), and mice subjected to donor contact at 6 dpi (6DC), and non-infected control mice ($n = 4-6$ /dpi). P-values were calculated using two-way ANOVA followed by Tukey's multiple comparisons test (A and B) (**** $P < 0.001$). Survival rate data was analyzed using the Kaplan–Meier method and compared using the log-rank test.

Figure 3,4

(ii) It would be great if you can mark the positive cells of any staining in the donor tissue section for the ease of understanding the histology.

Response: We thank the reviewer for this helpful suggestion. We have marked the positive cells in the donor tissue section using black arrows to facilitate interpretation and have modified the legend accordingly (Revision Figure 3, new Figure 3E, 3H, 4A, 4D, 4G and 4J in the manuscript, page 34, lines 743, 746–747 and page 35, lines 758–759).

Revision Figure 3

Revision Figure 3. IHC images of the N protein in mouse lungs from mice in the donor. Scale bars: 500 μm (top rows); 100 μm (bottom rows). Positive cells are indicated by black triangles. IHC images of N protein in mouse nasal conchae from mice in the donor. Scale bars: 200 μm (top rows); 50 μm (bottom rows). Black triangles indicate positive cells. (C-F) IHC images of the immune cells in mouse lungs from mice in the donor. (C) F4/80 (macrophage marker), (D) Anti-neutrophil (neutrophil marker), (E) CD3B (T cell marker), and (F) PTPRC (B cell marker) detection. Scale bars: 100 μm . Black triangles indicate positive cells.

Reviewer #2 (Comments for the Author):

Major points:

1) Line 152: The authors state that the viral load was the highest at 2 days post-infection (dpi) and declined by 7 dpi. However, were viral loads measured at other time points? It is possible that the peak occurred on day 3, which should be clarified.

Response: We thank the reviewer for the insightful comment. While this manuscript presents the viral load data at selected time points, we have previously conducted a separate experiment using the same intranasal infection dose (1×10^5 PFU) of SARS-CoV-2, in which viral titers in the lungs were measured at multiple time points. The results showed peak viral load at 2 dpi, followed by a gradual decline through 7 dpi (Revision Figure 4, page 8, lines 148 in the manuscript, and new Figure S2A in the Supplemental Material). These findings are consistent with those of previous studies reporting viral kinetics in the lungs following intranasal infection, which we have now cited in the revised manuscript to support the reliability of our data (1).

Revision Figure 4

Revision Figure 4. (A) Plaque assay of lung tissues from K18-hACE2 mice (C57BL/6 background) infected with 1×10^5 PFU of SARS-CoV-2 via intranasal administration. Lung tissues were collected at the indicated time points post-infection, and viral titers were determined using the plaque assay. Data are presented as PFU per lung. Statistical analysis was performed using one-way ANOVA, and significance was defined as *** $P < 0.005$ and **** $P < 0.001$.

2) Figure 2: Why was the co-housing experiment conducted for only 48 hours? A rationale for this timeframe should be provided.

Response: We thank the reviewer for this thoughtful comment. The 48-h co-housing period was deliberately selected to evaluate the transmissibility of SARS-CoV-2 at two distinct stages of infection in donor mice: the early phase (characterized by localized infection in the respiratory tract and oral cavity) and the late phase (involving systemic infection). Our aim was to determine whether contact transmission could be initiated during early localized viral shedding versus more advanced systemic infection.

Although previous studies have not specifically investigated contact transmission in mice, they have shown that peak viral shedding in the respiratory tract occurs within the first 2–3 days post-infection, aligning with the early transmission window (2, 3). Therefore, we limited the co-housing duration to 48 h to capture this critical period of potential exposure while minimizing confounding effects of prolonged housing, such as environmental contamination or indirect transmission routes. This design allowed us to more precisely examine the impact of direct contact at specific stages of infection.

3) The study uses a high infection dose of 10^5 PFU of SARS-CoV-2 in K18-hACE2 mice. What is the justification for administering such a high viral dose?

Response: We appreciate the reviewer's thoughtful comment. The primary aim of our study was to investigate whether contact transmission could occur even when donor mice were in a state of systemic infection, a stage at which the probability of transmission is expected to be maximized. Administering this high viral dose (10^5 PFU) ensured a robust and widespread infection in donor mice, including viral presence in the respiratory tract, oral cavity, and feces (4). Our approach was based on previous studies using K18-hACE2 mice, where this dose reliably promoted detectable viral shedding in both oral and fecal samples, thereby enhancing the probability of contact-based transmission.

4) Figure 3D: Was the viral titer measured from a donor at 2 dpi or 6 dpi? If it was 2 dpi, the viral loads appear much lower than in Figure 1E. Please clarify.

Response: We thank the reviewer for this insightful comment. The viral titers for the donor group were measured at 6 dpi, following the 48-h co-housing experiment. Lung tissues were collected either from donor mice either at the end of the end of the 48-h contact period or upon death during that period. As viral titer measurements were taken at a later stage of infection, viral loads were generally lower and showed greater inter-individual variability compared to the peak titers observed at 2 dpi in Figure 1E. Therefore, the timing of collection might account for this apparent difference in viral load levels.

5) Viral RNA levels should be measured using RT-qPCR in the co-housing experiments for more quantitative analysis.

Innate immune responses can also be assessed using RT-qPCR, as it provides a more sensitive and quantitative measure.

Response: We thank the reviewer for this suggestion. To provide a more quantitative analysis, we performed RT-qPCR on RNA extracted from lung tissues of both donor and contact groups. SARS-CoV-2 RNA targeting the surface glycoprotein gene was detected (Ct values < 25) in donor mice from both C57BL/6 and FVB/NJ background K18-hACE2 strains, indicating active infection. In contrast, all contact and negative control mice showed Ct values ≥ 38 or were undetermined, consistent with the absence of detectable viral RNA (Revision Figure 5A and 5E). These findings are in agreement with our IHC, *in situ* hybridization, and plaque assay results, further supporting that contact transmission did not occur under these experimental conditions. Additionally, to assess the innate immune response, we quantified the expression levels of *Cxcl10*, *Ifnb1*, and *Il6* in lung tissues using RT-qPCR. These genes were significantly upregulated in the donor group, while their expression levels in the contact group were comparable to those of uninfected controls (Revision Figure 5B-D and 5F-H). These findings further support the absence of productive infection in contact-exposed mice.

Revision Figure 5

Revision Figure 5. (A) Quantification of SARS-CoV-2 surface glycoprotein gene expression in lung tissues from C57BL/6 background mice following the co-housing experiment. RT-qPCR was performed on total RNA extracted from lung tissues of each group (ND: not detected; Ct > 40). Bars represent mean \pm SEM from biological replicates. (B–D) mRNA expression levels of *Cxcl10*, *Ifnb1*, and *Il6* in lung tissues from the same co-housing experiment in C57BL/6 background mice.

(E) Quantification of SARS-CoV-2 surface glycoprotein gene expression in lung tissues from FVB/NJ background mice subjected to the co-housing experiment with MMC treatment. RT-qPCR was performed on total RNA extracted from lung tissues of each group (ND: not detected; Ct > 40). Bars represent mean \pm SEM from biological replicates. (F-H) mRNA expression levels of *Cxcl10*, *Ifnb1*, and *Il6* in lung tissues from FVB/NJ background mice subjected to the co-housing experiment with MMC treatment.

6) Figure 5: The viral titer in donor lung tissue (Figure 5C) is unexpectedly low, despite infection with 10^5 PFU/ml of the virus. This raises concerns: if the FVB background K18-hACE2 mice are not efficiently infected, then the interpretation of the contact transmission experiment may be inherently limited.

Response: We thank the reviewer for raising this question. We acknowledge that the viral titer shown in Figure 5C appears lower than expected. Upon careful review, we realized that the lung PFU data presented for the FVB/NJ mice were not measured at the intended early time point (2 dpc), the most relevant assessing donor infectivity. Instead, the data reflect the viral PFU obtained from a single donor mouse that survived until 6 dpc following co-housing. Viral loads at this time point may not accurately represent the peak viral load in FVB/NJ donor mice during the early infectious window.

Due to constraints in animal availability during the revision period, we were unable to repeat the co-housing experiment or perform PFU measurements at the intended earlier time point (e.g., 2 or 3 dpi) in FVB/NJ mice. However, to address this limitation, we have included additional data from a separate experiment (now presented in Revision Figure 6), in which FVB/NJ K18-hACE2 mice were intranasally infected with 10^5 PFU of SARS-CoV-2, and lung viral titers were measured at 5 dpi. These data demonstrate that the viral PFU in FVB/NJ mice at 5 dpi are comparable to those observed in C57BL/6 mice infected under identical conditions, suggesting that the efficiency of viral replication is not significantly influenced by genetic background.

While we acknowledge the limitation in the dataset originally shown, the additional data support that FVB/NJ K18-hACE2 mice can be productively infected with SARS-CoV-2. Therefore, we believe that the findings from our contact transmission experiments remain valid.

Revision Figure 6

Revision Figure 6. (A) Quantification of infectious SARS-CoV-2 titers in lung tissues of C57BL/6 and FVB/NJ background K18-hACE2 mice at 5 days post intranasal infection with 10^5 PFU. Plaque assays were performed on lung homogenates to determine viral loads. Bars represent mean \pm SEM from biological replicates.

7) Figure S4B and S4D do not include donor body weight and temperature data. These should be provided.

Response: We thank the reviewer for pointing this out. We recognize the importance of these parameters for assessing the severity of infection in donor animals. Unfortunately, due to constraints in animal availability during the revision period, we were unable to repeat the experiment or obtain additional data from FVB/NJ K18-hACE2 donor mice.

To address this limitation, we reviewed previously published studies using the same FVB/NJ K18-hACE2 background and infection dose. These studies have reported a progressive loss of body weight starting around 3 dpi, suggesting active disease progression (5). Although direct comparisons between FVB/NJ and C57BL/6 backgrounds are limited, the observed patterns of weight loss in the FVB/NJ strain were generally comparable to those observed in our C57BL/6 donor group.

Although we were unable to measure clinical parameters such as weight or temperature in the FVB/NJ donor mice used in our experiment, the fact that mortality was only observed in the donor group and that infectious virus was detected in their respiratory tissues at the time of necropsy suggests successful infection. These findings collectively confirm that donor mice were effectively infected and validate the experimental design used to assess contact transmission.

8) Why was the co-housing duration reduced from 48 hours to 24 hours in the MMC-treated experiments? Please clarify the reasoning for this modification.

Response: We thank the reviewer for noting this important detail. Upon review, we identified a labeling error with respect to the co-housing duration in the MMC-treated experiments. We have corrected the figure accordingly in the revised manuscript (Revision Figure 7, new Figure 5A in the revised manuscript, page 35, lines 763–764).

To clarify, the contact transmission experiment using FVB/NJ background mice (including the MMC-treated groups) followed the same protocol as the C57BL/6 background experiment, with a 48-h co-housing period.

Revision Figure 7

Revision Figure 7. (A) Schematic image showing a model of contact transmission and induced immunodeficiency through MMC treatment. Donor and recipient mice were co-housed for 48 h before assessing transmission outcomes.

9) Statistical tests are missing for some figures.

Response: We thank the reviewer for this detailed review and for highlighting the missing information on statistical tests. Accordingly, we have completed the statistical analysis for the previously omitted datasets and have updated the relevant figures and legends in the revised manuscript to reflect changes (Revision Figure 1, new Figure 1A-C, Revision Figure 2, new Figure 2B-D, and Revision Figure 8, new Figure S5D in the Supplemental Material). All the newly inserted information is highlighted in yellow in the new version of the manuscript.

Revision Figure 8

Revision Figure 8. (A) Survival rates of FVB/NJ background K18-hACE2 mice intranasally infected with 1×10^5 PFUs of SARS-CoV-2 (Donor), recipient mice, MMC-treated recipient mice co-housed with donors, non-infected MMC-treated control mice, and negative-control mice. Survival rate data were analyzed using the Kaplan–Meier method and compared using the log-rank test.

Minor points:

1) Please clearly mention the variant of SARS-CoV-2 instead of the clade. The authors are using it interchangeably which is confusing for the readers.

Response: We appreciate the reviewer for this careful observation. As pointed out, the interchangeable use of "variant" and "clade" in the original manuscript may potentially be confusing to readers. We have revised the manuscript to consistently refer to all SARS-CoV-2 strains using the term "variant", which are now highlighted in yellow in the revised manuscript (page 6, line 115, page 12, lines 254-255, page 13, line 257, and page 16, line 331).

2) A 2020 (Bao et. al. JID) study reported contradictory results on the contact transmission of SARS-CoV-2 in K18-hACE2 mice. The authors should discuss this prior study in more detail and explain why their findings are different.

Response: We thank the reviewer for this insightful comment and the opportunity to clarify this important point. The discrepancy between our findings and those of Bao et al. (2020, JID) likely stems from the use of different transgenic mouse models expressing human ACE2 under distinct promoters.

Our study used the K18-hACE2 mouse model, in which hACE2 expression is driven by the human keratin-18 (K18) promoter. This promoter drives robust expression in the lower respiratory tract, particularly in alveolar epithelial cells, with a relatively limited expression in the upper airway epithelium, including the nasal and bronchial regions. Consequently, SARS-CoV-2 infection in K18-hACE2 mice predominantly involves the lungs, with minimal replication in the upper airways.

Supporting this, previous studies have shown that by 3 dpi, viral RNA levels in the lungs of K18-hACE2 mice can reach approximately 10^{10} copies/g, while levels in the nasal epithelium are up to 10^5 -fold lower (6). Similarly, Winkler et al. reported that infectious viral titers in the nasal turbinate were approximately 100-fold lower than those in the lungs at 2 dpi (1). These findings suggest that viral replication in the upper airway is limited in this model, and although low levels of viral RNA have been detected in nasal or oral swabs, they are likely insufficient to promote efficient contact transmission.

Furthermore, the rapid disease progression and high mortality observed in K18-hACE2 mice may shorten the window of viral shedding, further limiting opportunities for transmission.

In contrast, the transgenic hACE2 mouse model used by Bao et al. was generated using the endogenous mouse ACE2 promoter, which drives hACE2 expression in a pattern more consistent with native ACE2 distribution, including robust expression in the nasal and airway epithelium. This may have enabled higher viral replication in the upper respiratory tract, prolonged viral shedding, and consequently, successful contact transmission.

Therefore, we believe that the contrasting results between our study and that of Bao et al. are attributable to differences in hACE2 transgene regulation and tissue tropism, which are critical for viral transmission.

References

1. Winkler ES, Bailey AL, Kafai NM, Nair S, McCune BT, Yu J, Fox JM, Chen RE, Earnest JT, Keeler SP, Ritter JH, Kang LI, Dort S, Robichaud A, Head R, Holtzman MJ, Diamond MS. 2020. SARS-CoV-2 infection of human ACE2-transgenic mice causes severe lung inflammation and impaired function. *Nat Immunol* 21:1327–1335.
2. Kim YI, Kim SG, Kim SM, Kim EH, Park SJ, Yu KM, Chang JH, Kim EJ, Lee S, Casel MAB, Um J, Song MS, Jeong HW, Lai VD, Kim Y, Chin BS, Park JS, Chung KH, Foo SS, Poo H, Mo IP, Lee OJ, Webby RJ, Jung JU, Choi YK. 2020. Infection and rapid transmission of SARS-CoV-2 in ferrets. *Cell Host Microbe* 27:704–709.e2.
3. Munster VJ, Feldmann F, Williamson BN, van Doremalen N, Pérez-Pérez L, Schulz J, Meade-White K, Okumura A, Callison J, Brumbaugh B, Avanzato VA, Rosenke R, Hanley PW, Saturday G, Scott D, Fischer ER, de Wit E. 2020. Respiratory disease in rhesus macaques inoculated with SARS-CoV-2. *Nature* 585:268–272.
4. Yinda CK, Port JR, Bushmaker T, Offei Owusu I, Purushotham JN, Avanzato VA, Fischer RJ, Schulz JE, Holbrook MG, Hebner MJ, Rosenke R, Thomas T, Marzi A, Best SM, de Wit E, Shaia C, van Doremalen N, Munster VJ. 2021. K18-hACE2 mice develop respiratory disease resembling severe COVID-19. *PLOS Pathog* 17:e1009195.
5. Seo SM, Son JH, Lee JH, Kim NW, Yoo ES, Kang AR, Jang JY, On DI, Noh HA, Yun JW, Park JW, Choi KS, Lee HY, Shin JS, Seo JY, Nam KT, Lee H, Seong JK, Choi YK. 2022. Development of transgenic models susceptible and resistant to SARS-CoV-2 infection in FVB background mice. *PLOS One* 17:e0272019.
6. Oladunni FS, Park JG, Pino PA, Gonzalez O, Akhter A, Allué-Guardia A, Olmo-Fontánez A, Gautam S, Garcia-Vilanova A, Ye C, Chiem K, Headley C, Dwivedi V, Parodi LM, Alfson KJ, Staples HM, Schami A, Garcia JI, Whigham A, Platt RN, Gazi M, Martinez J, Chuba C, Earley S, Rodriguez OH, Mdaki SD, Kavelish KN, Escalona R, Hallam CRA, Christie C, Patterson JL, Anderson TJC, Carrion R, Dick EJ, Hall-Ursone S, Schlesinger LS, Alvarez X, Kaushal D, Giavedoni LD, Turner J, Martinez-Sobrido L, Torrelles JB. 2020. Lethality of SARS-CoV-2 infection in K18 human angiotensin-converting enzyme 2 transgenic mice. *Nat Commun* 11:6122.

Once again, we would like to express our sincere gratitude to the editors and reviewers for their positive and constructive criticism. The manuscript has vastly benefited from your valuable and insightful comments and suggestions. We look forward to hearing from you and would be happy to address any further concerns, if required. We hope this further pushes the manuscript closer to publication in your esteemed journal.

Re: Spectrum03413-24R1 (Adult K18-hACE2 Mice Are Suitable for Studying Intranasal SARS-CoV-2 Infection but Not Direct-Contact Transmission)

Dear Prof. Ki Taek Nam:

Thank you for the privilege of reviewing your work. Below you will find my comments, instructions from the Spectrum editorial office, and the reviewer comments.

Please, incorporate the modifications as suggested by reviewer #2.

Revision Guidelines

Sincerely,
Luciana Costa
Editor
Microbiology Spectrum

Reviewer #1 (Comments for the Author):

I would like to thank the authors for addressing all my comments.

Reviewer #2 (Comments for the Author):

The authors have addressed all of my concerns. However, I recommend that they include Revision Figure 5 in the manuscript (supplementary). Additionally, please include the explanation stated in the response for the rationale for selecting the 48 hr timeframe for the co-housing experiment in the manuscript. Finally, the discussion section should include the explanation for the discrepancy between this study's findings and those of Bao et al. The authors have explained the likely reason for this difference in their response; including this explanation in the discussion would strengthen the manuscript.

Reviewer #2 (Comments for the Author):

The authors have addressed all of my concerns. However, I recommend that they include Revision Figure 5 in the manuscript (supplementary). Additionally, please include the explanation stated in the response for the rationale for selecting the 48 hr timeframe for the co-housing experiment in the manuscript. Finally, the discussion section should include the explanation for the discrepancy between this study's findings and those of Bao et al. The authors have explained the likely reason for this difference in their response; including this explanation in the discussion would strengthen the manuscript.

Response: We thank the reviewer for this suggestion. In accordance with the reviewer's suggestions, we have incorporated RT-qPCR data for SARS-CoV-2 viral RNA levels as well as RT-qPCR data for cytokines associated with the innate immune response into the main text (new Supplemental figure 5 and Supplemental figure 6 in the manuscript, page 10, lines 190–194, page 11, lines 226–229, page 12, line 249, and page 13, lines 257–259). These additions further substantiate our conclusion that no detectable response occurred following contact transmission. We have also included a detailed rationale for selecting the 48-hour co-housing period, thereby providing a clearer justification for our experimental design (page 9, lines 173–177). Finally, we have integrated into the discussion section our explanation for the observed discrepancy between our findings and those reported by Bao et al., with the aim of strengthening the interpretation of our results and clarifying the differences relative to previous studies (page 15–16, lines 319–341, and lines 344–346). We sincerely appreciate the reviewer's constructive comments, which have significantly improved the quality and clarity of our manuscript.

Re: Spectrum03413-24R2 (Adult K18-hACE2 Mice Are Suitable for Studying Intranasal SARS-CoV-2 Infection but Not Direct-Contact Transmission)

Dear Prof. Ki Taek Nam:

Your manuscript has been accepted, and I am forwarding it to the ASM production staff for publication. Your paper will first be checked to make sure all elements meet the technical requirements. ASM staff will contact you if anything needs to be revised before copyediting and production can begin. Otherwise, you will be notified when your proofs are ready to be viewed.

Sincerely,
Luciana Costa
Editor
Microbiology Spectrum